# Design of a Variable-Stiffness Compliant Skin for a Morphing Leading Edge

**Zhigang Wang *** and **Yu Yang**

Aircraft Strength Research Institute of China, Xi'an 710065, China; yangyu@cae.ac.cn
* Correspondence: wangzhigang@cae.ac.cn

**Featured Application: This research focuses on the morphing leading edge design, which can be applied in a morphing wing for future civil aircraft.**

**Abstract:** A seamless and smooth morphing leading edge has remarkable potential for noise abatement and drag reduction of civil aircraft. Variable-stiffness compliant skin based on tailored composite laminate is a concept with great potential for morphing leading edge, but the currently proposed methods have difficulty in taking the manufacturing constraints or layup sequence into account during the optimization process. This paper proposes an innovative two-step design method for a variable-stiffness compliant skin of a morphing leading edge, which includes layup optimization and layup adjustment. The combination of these two steps can not only improve the deformation accuracy of the final profile of the compliant skin but also easily and effectively determine the layup sequence of the composite layup. With the design framework, an optimization model is created for a variable-stiffness compliant skin, and an adjustment method for its layups is presented. Finally, the deformed profiles between the directly optimized layups and the adjusted ones are compared to verify its morphing ability and accuracy. The final results demonstrate that the obtained deforming ability and accuracy are suitable for a large-scale aircraft wing.

**Keywords:** morphing; leading edge; optimization; variable stiffness; composite



## 1. Introduction

China is becoming one of the world's largest aviation markets, whose fleet, according to Boeing's Commercial Market Outlook 2019–2038 [1], will account for nearly 20% of the share of the world by 2037. To meet the strict requirements of green aviation, China has been actively encouraging aircraft designers to develop innovative technologies [2]. Seamless and smooth morphing leading edge has been one of the most potential aviation technologies in the short-term future from the perspectives of noise abatement and drag reduction [3–5]. Firstly, the gap and slot introduced by conventional leading edge produce a large part of noise during takeoff and landing, and eliminating them is a good cause [2,6]. Secondly, the morphing leading edge is detrimental to laminar flow, which requires the skin surface quality to an extremely high extent [7,8]. Therefore, developing such a new system has a remarkable advantage in noise abatement and drag reduction.

Much work has been carried out for this purpose [3,9–24], and such work can be roughly divided into two types, depending on whether it adopted a uniform-stiffness compliant skin or a variable-stiffness one. Due to good deformation accuracy and scalability, the latter is the concept with the most potential. The core issue, however, for this kind of concept is the optimization of the stiffness distribution of the complaint skin. To address this issue, Kintscher firstly employed an empirical design method to determine the stiffness distribution of a composite laminate compliant skin [19] and, in 2011, adopted a Simplex method to further optimize its thickness distribution [21]. With a similar design method, Rose et al. designed a morphing leading edge with a variable compliant skin and an

internal compliant structure in 2014 [23]. Based on homogeneous material and a shape optimization method, Cavalieri et al. designed a variable-thickness compliant skin [9]. Thuwis et al. chose equivalent axial and bending stiffness as a design variable to optimize the distribution of a composite compliant skin, without any consideration of the actual layup sequence [12]. In short, to realize a variable stiffness concept, the previous research studies either adopted a variable-thickness homogeneous-material complaint skin or a variable-stiffness composite one. For the homogeneous-material complaint skin concept, it is difficult to obtain high deformation accuracy, as the search space of the variable is lower than the composite one. However, for the variable-stiffness composite one, the proposed design methods in the previous works cannot take manufacturing constraints or layup sequence of the composite laminate into account, as they either empirically determined it or did not care about them at all, which could cause high deviation of the final deformed profile from the aerodynamic target profile. In addition, the employed optimizer of these works was mainly a gradient-based method or the Simplex method, which is prone to be trapped in a locally optimal solution, decreasing the deformation accuracy further.

This paper, therefore, proposes an innovative two-step design method, including layup optimization and layup adjustment, for a variable-stiffness compliant skin based on composite material. The combination of these two steps can not only improve the deformation accuracy of the final profile of the compliant skin but also easily and effectively determine the sequence of the composite layups at the same time.

## 2. Definition of Optimal Aerodynamic Profile

The Chinese Aeronautical Establishment (CAE) has initiated a National Research Project titled "Variable Camber Wing Technology (VCAN)" for the next generation of long-range civil aircraft [25]. In the VCAN program, a morphing wing with adaptive leading and trailing edge is anticipated to be used in a long-haul business jet with a cruise Mach number of 0.87 [20]. It comprises a low-mounted backswept wing with a high aspect ratio, two aft-fuselage mounted engines, and a T-tail (Figure 1). The overall length is 33 m, and the wingspan is 33.5 m including the winglets. The aircraft targets a range of 11,000–13,000 km with a cruise altitude of 43,000 feet [25].

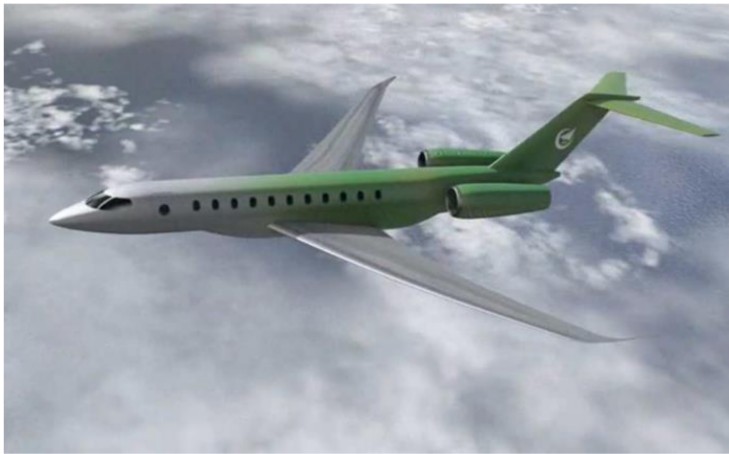

**Figure 1.** Artist impression of the long-haul business jet concept with cruise Mach number of 0.87 [25].

Before a structural design process, the first thing is to define the geometry input and geometry output. For a transport aircraft, the most common flight condition is the cruising case. It is reasonable to define the optimal aerodynamic surface in the cruising case as the initial profile, which will produce an easier actuation system and decrease the needed power, as the system can hold the cruising profile through a simple mechanical self-locking device in practical engineering. Inversely, the optimal aerodynamic surface in the takeoff or landing case is not the common case, but it is the most deflected one, which means the needed actuating power is the highest. Thus, the surface is defined as the target profile.

In the project, the optimal initial aerodynamic profile is provided by Hua et al., specifically in [26,27], and has been verified by a CAE-AVM wind tunnel test conducted in the DNWHST, a continuous closed-circle pressurized transonic wind tunnel. The basic profile for the wing design is NPU-SP6, which is a 13% thick supercritical airfoil [28]. Both inverse method [27] and numerical optimization [29] are applied to the wing design. The free-form deformation (FFD) approach is used for wing optimization. The CFD-estimated maximum lift to drag ratio L/D reaches 20 in cruise condition [30]. During the optimization, a CAE in-house code AVICFD-Y [31] is employed for the flow analysis and aerodynamic performance estimation. It is a Reynolds-averaged Navier–Stokes (RANS) solver based on multi-block structured mesh capable of large-scale flow analysis on parallel computer clusters. Typical grid sizes of 20–40 million nodes and the shear stress transport (SST) turbulence model are used, while other turbulence models and codes are also applied for comparison in design stage.

During the optimization for the target profile, the optimal initial profile obtained previously is utilized as the basic profile. The design variables are the leading edge radius and deflection angle, and the optimization objective is to maximize the lift coefficient. As there are only two design variables, a simple engineering parameter analysis method is used to determine the optimal target profile empirically. The variation range of the leading edge radius is set as 12, 14, 15, 16, and 18 mm, and the deflection angle is set to 20, 25, 30, 35, and 40°. Figure 2 presents the different leading edge profiles in the deflection angles of 20, 25, 30, 35, and 40° with a leading edge radius of 14 mm. Figure 3 shows the corresponding curves of the lift coefficient versus deflection angle. It can be seen that as the deflection angle increases, the maximum lift coefficient increases nonlinearly and reaches around 3.5. Once the deflection angle increases beyond 25°, the maximum lift coefficient increases at a slower rate. However, the circumferential length of the profiles in Figure 2 varies, which means the subsequent compliant skin needs to be extended or shortened during the deflection process. This is impossible for conventional metal or composite material in the aviation industry. Therefore, an additional constraint of a constant length should be included in the analysis process.

Finally, with the constant length constraint, a series of profiles is obtained, along with the corresponding maximum lift coefficient. For the selected aircraft, the necessary maximum lift coefficient during takeoff is far below the value of 3.5 shown previously. Therefore, a drooped profile with a deflection angle of 20° and a leading edge radius of 16 mm is used as the target profile because the corresponding maximum lift coefficient is sufficient for takeoff. The purpose of the present work is to establish a methodology to design a compliant skin for the morphing leading edge.

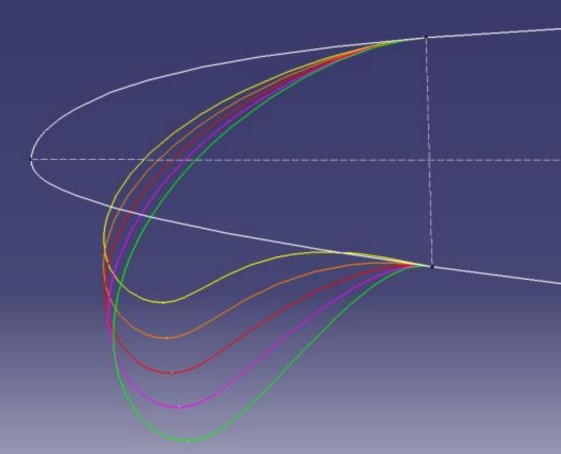

**Figure 2.** Different leading edge profiles in the deflection angles of 20, 25, 30, 35, and 40° with a leading edge radius of 14 mm.

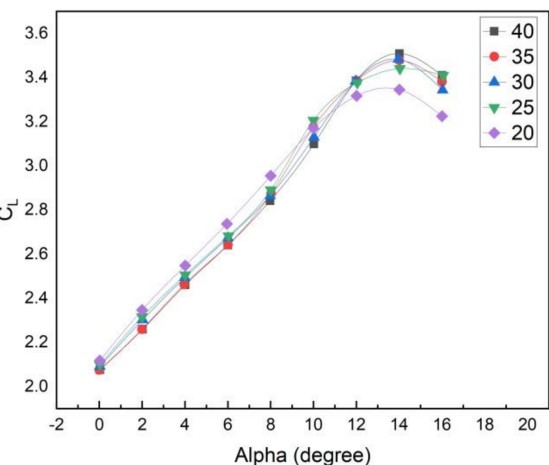

**Figure 3.** Curves of the lift coefficient versus deflection angle.

## 3. Structural Concept of Morphing Leading Edge

In the VCAN program, the design objective is a two-dimensional morphing wing physical mockup with a span size of 3 m, which will be used for ground and wind tunnel tests. Figure 4 presents the digital mockup concept of the morphing wing. The mockup integrates a morphing leading edge and a morphing trailing edge to be drooped seamlessly, smoothly, and precisely to an aerodynamic target profile. Due to the scalability of the concept and a lower budget of the physical mock-up manufacturing, this study extracts a morphing leading edge with a spanwise length of 350 mm as the design object, including only one set of internal actuating mechanism.

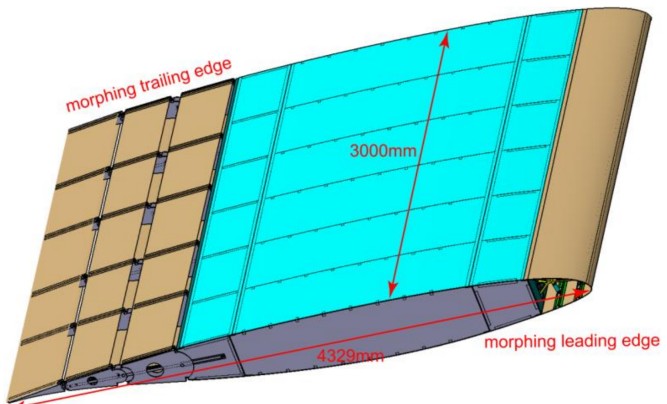

**Figure 4.** Two-dimensional morphing wing digital mockup with morphing leading and trailing edge.

As shown in Figure 5, the structural concept of the morphing leading edge is inspired by the research of Kintscher et al. [28], which also includes a compliant skin and an internal kinematic mechanism. The flexible skin is connected to a front spar to form a seamless and smooth profile. Four stringers are integrated to transfer the aerodynamic load to inner kinematic mechanisms. The mechanism is used not only to deflect the entire compliant skin but also to withstand the outer aerodynamic force. Obviously, the final deflected profile of the morphing leading edge is determined by the stiffness distribution of the compliant skin and the position of the stringers according to the classical spline theory. If the end-to-end compliant skin is constant in stiffness, the drooped profile will be uniquely determined by the interface displacement transferred by the internal mechanism at the four interface points, which seem like four control points of a spline curve. Therefore, it is necessary to use a variable-stiffness compliant skin and design it elaborately to obtain an aerodynamically optimal profile. The following two sections present the proposed

two-step design method and illustrate how it takes the manufacturing constraints into account collaboratively.

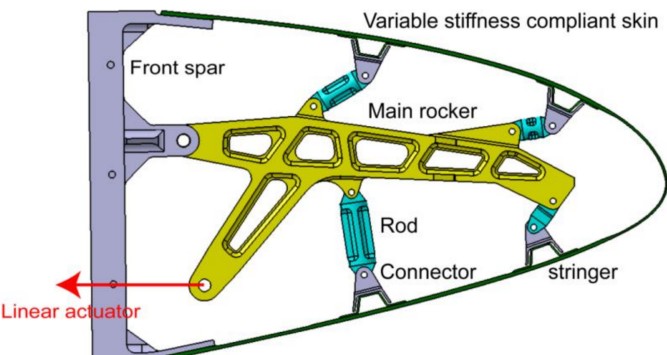

**Figure 5.** Structural concept of the morphing leading edge.

## 4. Optimization Method

### 4.1. FEM Model

To decrease material strain level, a pure bending principle is used here, which means there is no compression or stretching in the middle plane of the compliant skin, which still suffers severe bending strain. Therefore, this study employs an aviation-level woven glass fiber reinforced polymer (SW100A/6511, Weihai Guangwei Composites Co., Ltd., Weihai, China), which is characterized by high ultimate strain. The mechanical properties of this material are presented in Table 1.

**Table 1.** Mechanical properties of the woven glass fiber reinforced polymer laminate.

| Property Parameter | Value |
| --- | --- |
| Elastic modulus along fiber direction (GPa) | 23.20 |
| Elastic modulus transverse to fiber direction (GPa) | 23.20 |
| Poisson's ratio | 0.12 |
| Shear modulus (GPa) | 2.97 |
| Tensile strength along fiber direction (MPa) | 477.00 |
| Compressive strength along fiber direction (MPa) | 302.00 |
| Tensile strength transverse to fiber orientation (MPa) | 477.00 |
| Compressive strength transverse to fiber orientation (MPa) | 302.00 |
| Shear strength (MPa) | 54.00 |
| Tensile ultimate strain transverse to fiber orientation ($\mu$) | 33,166 |
| Compressive ultimate strain transverse to fiber orientation ($\mu$) | 13,538 |

Considering the complexity of the manufacturing process, the variable stiffness is realized by changing layup thickness and ply orientation of straight fiber layups in different regions. Taking the manufacturing realizability and deformation accuracy into account, the compliant skin is partitioned as 10 uniformly spaced thickness regions along the circumferential direction as shown in Figure 6. In each FEM analysis called for by the main optimization procedure, the internal mechanism can be simulated with actuated forces at the four respective interface points during optimization. As the skin deformation depends on the layup and the magnitude of the actuated forces, the layup and the forces all need to be optimized and defined as design variables. However, the variation range of the actuated forces is not certain in advance. Therefore, a series of trial calculations are needed to obtain the approximate ranges of the actuated forces, which are utilized as the optimization constraints and presented as Equation (5) in Section 4.2. Once the design variables of the actuated forces are given in every optimization step, the force boundary in every single analysis is certain. It is worth mentioning that a nonlinear finite element analysis, rather than a linear one, is employed to help produce a movement locus of the interface point above each stringer hat during the morphing process, which strongly supports later inner kinematic design.

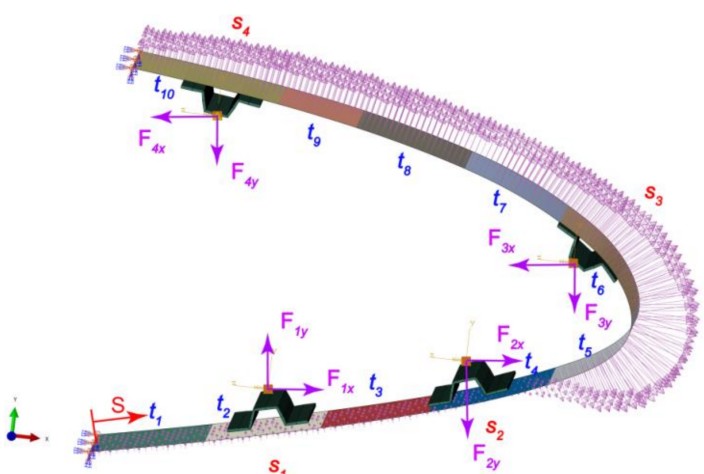

**Figure 6.** Finite element model of the variable-stiffness compliant skin.

Figure 6 presents the finite element model of the variable-stiffness compliant skin. In this study, the nonlinear FEM solver in Abaqus of Dassault Systems is used and called by a python script, which is an independent module for the main optimization procedure. For the nonlinear FEM solver, an automatic type of step increment is adopted with the maximum number of increments of 100. The initial, minimum, and maximum increment sizes are set as 1, $1 \times 10^{-5}$, and 1, respectively. The full Newton method is used as the solution technique for the FEM model. Finally, the solver gives the coordinates of every single node on the complaint skin, which are used to calculate the objective function of the compliant skin optimization problem. The nonlinear finite element analysis illustrated in this study is referred to as a geometrically nonlinear analysis. Therefore, the magnitude of the enforced forces is constant, while their direction is always changing, which means that only the direction of the distributed pressure should always be updated. In this study, an implicit structural solution is obtained to solve the geometrical nonlinear problem with an incremental approach of loading and an upgrading stiffness matrix accordingly. During the process, the direction and value of the pressure on the compliant skin are updated automatically.

*4.2. Optimization Model*

As illustrated before, the design objective for the morphing leading edge is to obtain the target profile as precisely as possible (Figure 7). To describe the deviation between the realized final profile and the target profile, unlike what was used in previous work [21,22], an improved objective function based on weighted least square error (WLSE), shown in Equation (1), is proposed for more delicate deviation control. The previous least square error (LSE) may be insufficient since the final LSE can be low enough even when unacceptable deformation deviation in a certain aerodynamically critical regime, such as the frontmost tip, still exists. Therefore, a weighted penalty is introduced to the LSE formula.

$$WLSE = \sum_{i=1}^{n} w_i \frac{d_i}{n} = \sum_{i=1}^{n} w_i \frac{\sqrt{\left(x_i - x_i^*\right)^2 + \left(y_i - y_i^*\right)^2}}{n} \tag{1}$$

In the above, $n$ is the number of monitoring points; $w_i$, $d_i$, $(x_i, y_i)$, and $(x_i^*, y_i^*)$ are the weight, the deviating distance from the target, the actual coordinate, and the target coordinate of the $i$th monitoring point, respectively. The weight coefficient follows a bi-linear law that is empirically generalized in Equation (2) over $n = 20$ uniformly spaced monitoring points. In Equation (2), s is the normalized curve length along the circumferential direction of the profile.

$$\begin{cases} w = 0.3 + 0.66667s \ for \ 0 < s \le 0.6 \\ w = 1.3 - s \ for \ 0.6 < s \le 1 \end{cases} \tag{2}$$

The design variables include layup sequence in each design zone, each stringer's position, and actuating force at each interface point. To simplify the optimization process, all design zones are assigned with the same layup pattern $[0_a/45_b/90_c/\text{-}45_d/90_e]s$, while virtual thicknesses a, b, c, d, and e vary from one zone to another ($0°$ degree aligns with the circumferential direction of the airfoil and $90°$ aligns with the spanwise direction). The constraint of variables a to e is that each of them must be an integral multiple of a single sheet. Obviously, this layup sequence is not fit for manufacturing, and a layup adjustment process is essential (discussed in Section 3). Incidentally, the layup sequence of four stringers is set uniformly as $[0/45/90/\text{-}45/\overline{90}]s$. Consequently, there are total 62 design variables: five variables defining layup sequence in each of the 10 design zones, the position of four stringers in terms of normalized length, and actuating force components in x and y directions in each of the four actuating points.

Furthermore, based on the pure bending principle, the strain in the skin results from pure bending, and the relationship between the maximum strain at the outmost skin surface and the curvature and thickness of the skin at that point is given in Equation (3):

$$t(s) = \frac{\varepsilon_{\lim}}{1/2\Delta\kappa(s)} \tag{3}$$

where $\varepsilon_{\lim}$ is the material limit in terms of strain, $s$ is defined as the normalized length along the skin profile from the lower end to the upper, $t$ is the maximum allowable thickness, and $\Delta\kappa$ is the curvature variation. Both $t$ and $\Delta\kappa$ are a function of $s$. According to the magnitude of the curvature variation between the clean case and drooped case, the local maximum skin thickness is determined from Equation (3) and then used as one of the constraints in the following skin stiffness distribution optimization process. Consequently, as for the selected GFRP, the maximum skin thickness around the circumferential length of 42% and 50%, where the greatest curvature variation occurs, must be less than 3.6 mm. This is used as one of the optimization constraints.

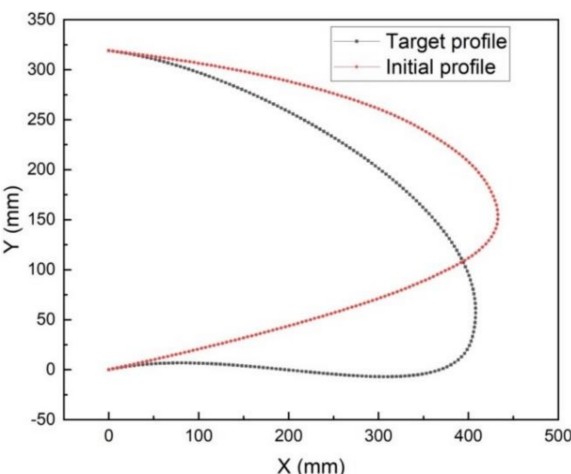

**Figure 7.** Initial profile and target profile.

Finally, the mathematical formulation of the variable-stiffness compliant skin optimization problem is defined in Equation (4):

$$
\begin{aligned}
\min WLSE &= \sum_{i=1}^{n} w_i \frac{d_i}{n} = \sum_{i=1}^{n} w_i \frac{\sqrt{(x_i - x_i^*)^2 + (y_i - y_i^*)^2}}{n} \\
s.t.: \quad & \mathrm{K}(\mathrm{t})\mathrm{U}(\mathrm{t}) = \mathrm{F}(\mathrm{t}) \\
& s_i^L < s_i < s_i^L, i = 1, \ldots, 4 \\
& f_{xi}^L < f_{xi} < f_{xi}^U, \ f_{yi}^L < f_{yi} < f_{yi}^U, i = 1, \ldots, 4 \\
& t_{ij}^L < t_{ij} < t_{ij}^U, i = 1, \ldots, 10, j = 1, \ldots, 5
\end{aligned}
\tag{4}
$$

where $K(t)$ is the stiffness matrix of the whole FEM model; $U(t)$ and $F(t)$ are its node displacement vector and node force vector induced by geometry nonlinearity, respectively; t is the virtual time during geometrically nonlinear analysis; $s_i$ is the position along the skin profile of the *i*th stringer in terms of normalized length; $s_i^U$ and $s_i^L$ are the upper and lower position limits of the *i*th stringer; and $f_{xi}$ and $f_{yi}$ are actuating loads exerted on the *i*th stringer in x and y directions, respectively. $(f_{xi}^L, f_{xi}^U)$ and $(f_{yi}^L, f_{yi}^U)$ are load limits of the *i*th stringer in x and y directions, respectively. Here, the load limits are given in Equation (5) based on several computing attempts:

$$
\begin{aligned}
f_{xi}^L &= -500\text{N},\ f_{xi}^U = 1800\text{N} \\
f_{xi}^L &= 1000\text{N},\ f_{xi}^U = 3500\text{N} \\
f_{xi}^L &= 1000\text{N},\ f_{xi}^U = 3000\text{N} \\
f_{xi}^L &= -1000\text{N},\ f_{xi}^U = 2500\text{N} \\
f_{yi}^L &= 200\text{N},\ f_{yi}^U = 2000N \\
f_{yi}^L &= -2500\text{N},\ f_{yi}^U = -500N \\
f_{yi}^L &= -1500\text{N},\ f_{yi}^U = 100N \\
f_{yi}^L &= -600\text{N},\ f_{yi}^U = 500N \\
t_{ij}^L &= 0.2,\ t_{ij}^U = 0.4
\end{aligned}
\tag{5}
$$

### 4.3. Optimization Algorithm

It is known that conventional gradient-based optimizers are not only inadequate for the current situation, which requires collaborative optimization involving both continuous variables (the actuating loads) and discrete ones (the stiffness distribution over the entire flexible skin, the composite layup sequence, and the actuating positions), but also likely to be trapped in locally optimal solutions. In contrast, the secondary generation of Non-dominated Sorting Genetic Algorithm (NSGA-II) [32] is known for its high efficiency and stability in terms of globally optimal solution search. Therefore, NSGA-II is employed here, and the optimization framework of the entire procedure is illustrated in Figure 8. At the beginning, when a generation of a population of design variables (namely individuals) has been initialized, the main procedure will call the finite element module to generate the same amount of numerical models of morphing leading edge, each of which is customized with the corresponding design variables. All these models are analyzed to simulate the drooping deformation, and its results are evaluated in terms of WLSE (namely the fitness valuation). Afterward, the main procedure updates these design variables according to respective evaluation results, such as selection, crossover, and mutation, and then generates the offspring design variables. So long as one loop is finished, the same procedure above will be repeated until the convergence criterion is met.

In the algorithm, there are 62 design variables in total: five variables defining layup sequence in each of the 10 design zones, the position of four stringers in terms of normalized length, and the actuating force components in x and y directions in each of four actuating points. According to the recommendation from Reference [32] that the population should be 4 times larger than the design variable quantity, the population here is set as 200. When variations of the optimal and average value in two consecutive generations are both less than $1.0 \times 10^{-6}$, convergence is regarded to be achieved.

As illustrated before, there exists a large deformation of the complaint skin during the drooping process, which produces a geometrically linear problem. Specifically, the external aerodynamic force direction in the structural FEM model should change in real time to remain perpendicular to the skin surface. Therefore, a nonlinear finite element analysis (within the right-hand side frame of Figure 8), rather than the linear one, has been employed in the FEM analysis, which will also help to produce movement locus of the actuating points on stringers along optimizing process. Actually, it strongly supports later inner kinematic mechanism design, although it takes relatively longer computational time.

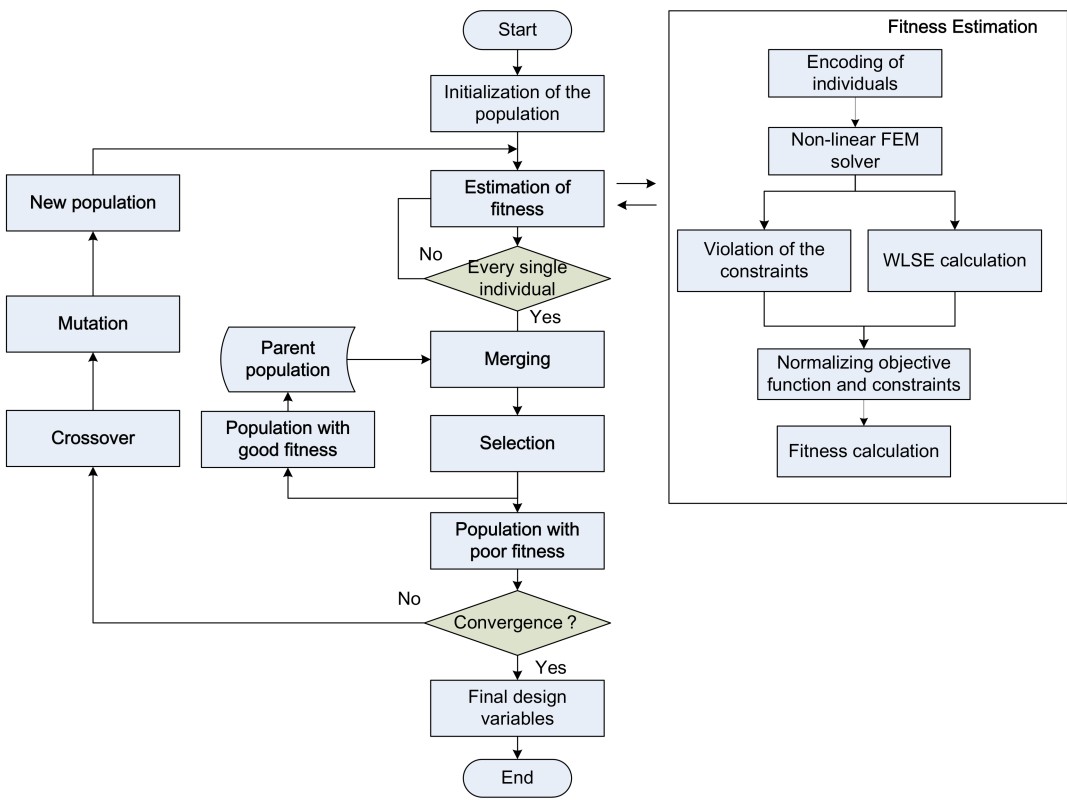

**Figure 8.** Optimization framework of the compliant skin based on NSGA-II.

### 4.4. Optimization Results

As shown in Figure 9, after 101 optimizing iterations, the converged result is obtained. The final WLSE is 1.1845 mm, which means the average distance deviation at all monitoring points from the target is 1.1845 mm. The final obtained profile is compared with the target one, which shows great agreement in Figure 10. Figure 11 presents the corresponding deviation distribution in the circumferential direction, and it can be seen that the maximum deviation is about 4.5 mm. This error is negligible when compared with the overall size of the morphing leading edge (about 450 mm in chord length).

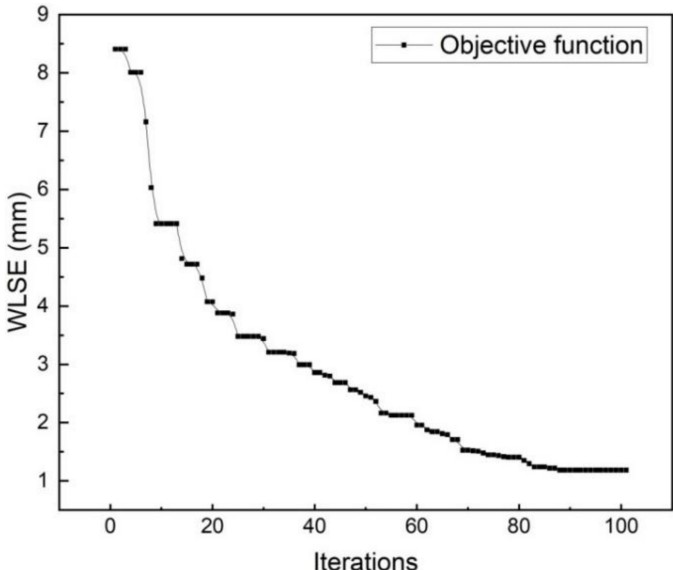

**Figure 9.** Convergence process of the optimization iterations.

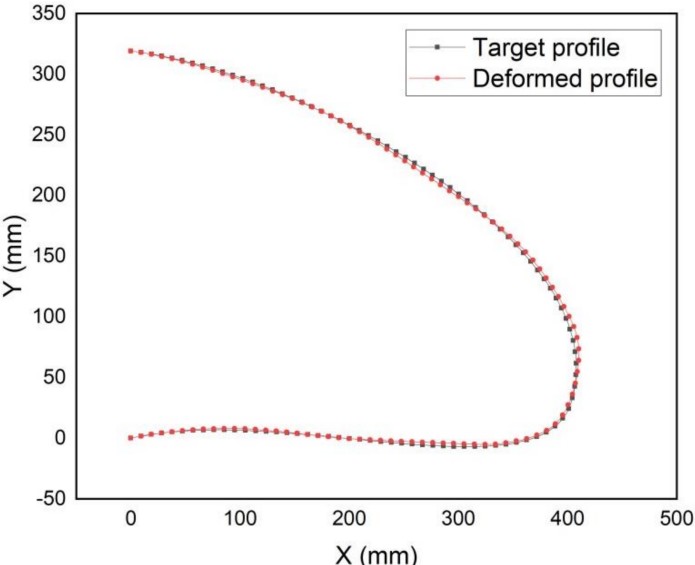

**Figure 10.** Comparison between target profile and deformed profile.

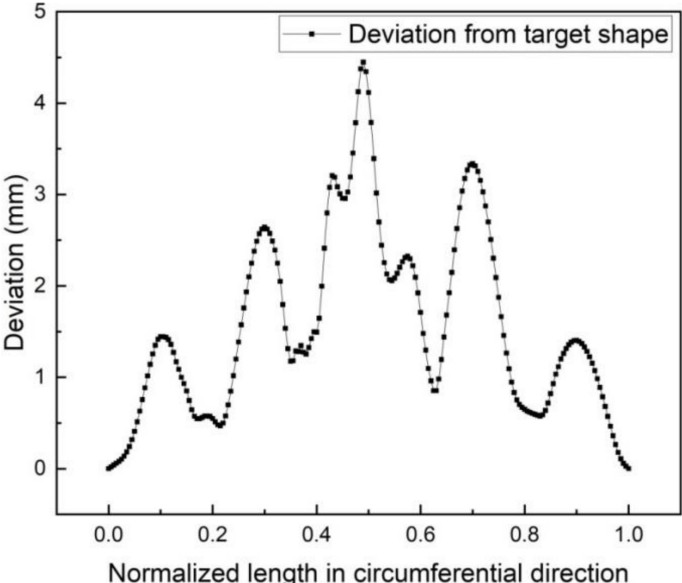

**Figure 11.** Deviation distributions along normalized length in circumferential direction.

## 5. Layup Adjustment Method

The primitive layup sequence obtained through optimization is presented in Table 2. Obviously, it is not fit for laminate manufacturing, as it not only violates the basic principle of stacking no more than three plies together along the same direction but also has a step thickness variation between two adjacent thickness regions that is detrimental to its strength. Therefore, a corresponding adjustment is essential, as seen in the engineering adjustment shown in Figure 12.

**Table 2.** Primitive and adjusted layup sequences.

| Zone | Primitive Layup | Updated Layup | Thickness (mm) |
|---|---|---|---|
| 1 | $[0_3/45_4/90_3/-45_3/90]s$ | $[(45/0)_7]s$ | 2.8 |
| 2 | $[0_4/45_3/90_4/-45_4/90_2]s$ | $[(45/0)_6/45(45/0)_2]s$ | 3.4 |
| 3 | $[0_4/45_4/90_4/-45_4/90_2]s$ | $[(45/0)_9]s$ | 3.6 |
| 4 | $[0_3/45_4/90_3/-45_4/90_2]s$ | $[(45/0)_8]s$ | 3.2 |
| 5 | $[0_2/45_2/90_2/-45_2/90]s$ | $[(45/0)_2/0/45/45/0/45]s$ | 1.8 |
| 6 | $[0_2/45_3/90_2/-45_2/90_3]s$ | $[(45/0)_2/(0/45)_30]s$ | 2.1 |
| 7 | $[0_4/45_4/90_4/-45_4/90_2]s$ | $[(45/0)_9]s$ | 3.6 |
| 8 | $[0_4/45_4/90_4/-45_4/90_2]s$ | $[(45/0)_9]s$ | 3.6 |
| 9 | $[0_4/45_4/90_4/-45_4/90_2]s$ | $[(45/0)_9]s$ | 3.6 |
| 10 | $[0_3/45_4/90_4/-45_2/90_2]s$ | $[(45/0)_4/45/(45/0)_3]s$ | 3.0 |

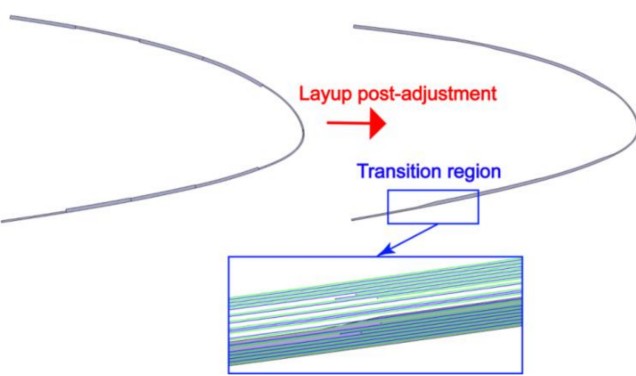

**Figure 12.** Engineering adjustment of the variable-thickness laminate including transition region.

Given this, several adjustment principles complying with manufacturing constraints were used in this study, as illustrated in detail as follows:

- Employing a symmetry and balance layup to the greatest extent to avoid warping produced by stiffness coupling;
- Avoiding a layup including two consecutive sheets with the same direction and otherwise ensuring that the number of the sheets with the same direction is lower than four;
- Applying a slope transition pattern symmetrically between two adjacent thickness regions with a slope of 1/10 as shown in Figures 13 and 14;
- If possible, ensuring that the length of a constant thickness region is larger than the minimum value of the length of the two neighboring transition regions when continuous transition regions occur (as shown in Figure 15, the length of C should be longer than the minimum lengths of A and B);
- Employing the preferred pattern shown in Figure 16 for transition regions (the second pattern is preferred as it can reduce the impact of stress concentration);
- Reserving a compensating distance of 3–5 mm for assembling convenience, as shown in Figure 17.

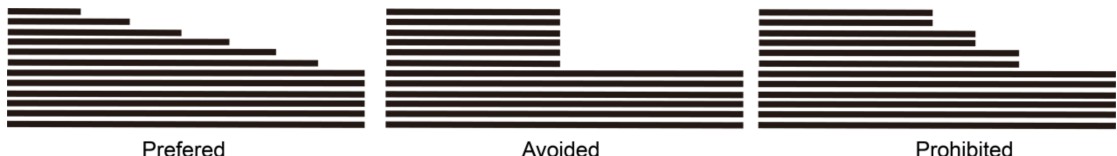

Prefered     Avoided     Prohibited

**Figure 13.** Different transition patterns.

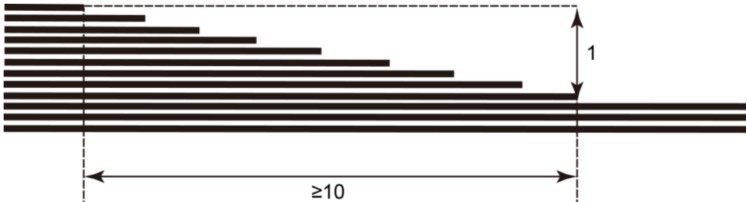

**Figure 14.** Recommended slope of transition region.

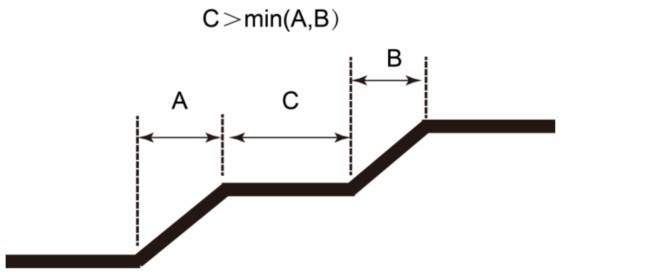

**Figure 15.** Requirement for continuous transition regions.

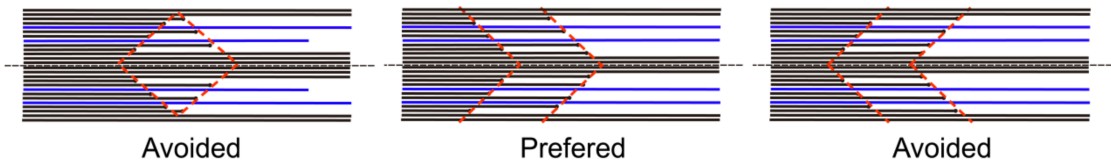

**Figure 16.** Patterns of the different kinds of transition regions.

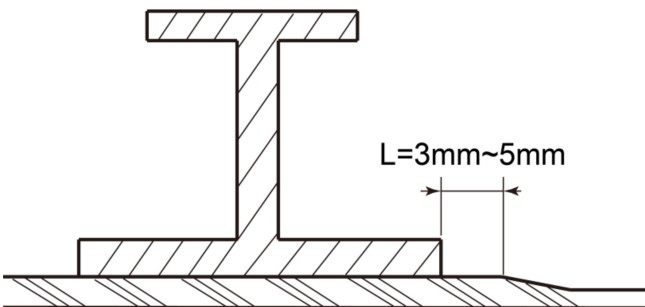

**Figure 17.** Reserved compensating distance between the transition region and the stiffener.

After employing these principles, the layup of the compliant skin was more engineeringly practical (Table 2). However, it needs to be emphasized that in this case not only manufacturability but also the variation of bending stiffness (BS) of skin needs to be taken into account because the morphing effect of the leading edge is realized through skin bending and the variation of BS along the skin plays a significant role in morphing effect control. As the adjustment of layup sequence can influence the BS somewhat inevitably, such impact should be minimized as much as possible. The adjustment is presented in Table 2.

## 6. Analysis of Results

### 6.1. Shape Accuracy

The final impact of the engineering adjustment of the layups to the profile deviation is analyzed in this section. Figures 18 and 19 show the comparisons of the final profile between the primitive layup (without engineering adjustment) and the adjusted layup (with engineering adjustment) in the drooped case. It can be seen that the maximum additional deviation introduced by the engineering adjustment is about 0.4 mm, which is almost negligible. Notably, the proposed layup adjustment method does not introduce

obvious accuracy error. This demonstrates that the proposed adjustment method can ensure high shape accuracy and meet the demands of manufacturing simultaneously. From Figure 19, it can also be seen that the maximum deviation is located around s = 0.5, which may result from the higher curvature variation in this region.

However, to decrease the impact of the engineering adjustment further, the optimization method should be improved in future work, such as by adopting a single-step optimization method directly embedded with manufacturing constraints.

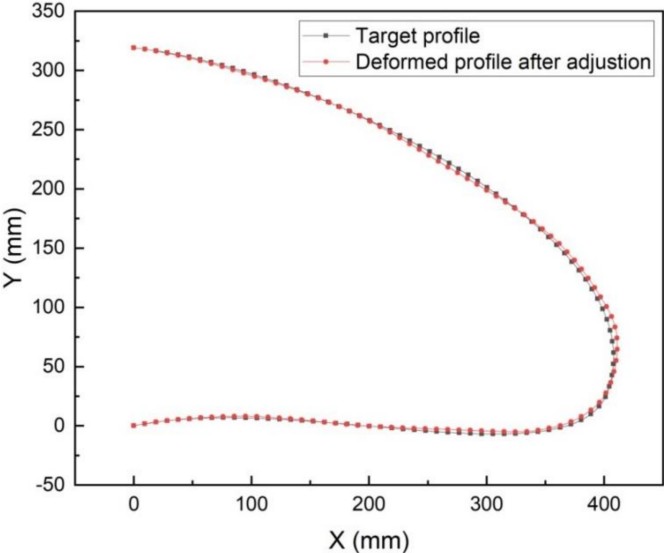

**Figure 18.** Comparison of profiles between the primitive and adjusted layup in drooped case.

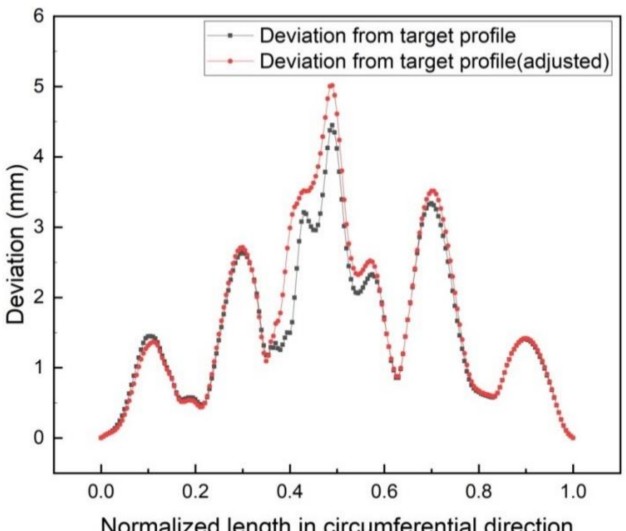

**Figure 19.** Deviation comparison between the primitive and adjusted layup in drooped case.

Besides the shape accuracy in drooped (or take-off) flight condition, the cruising condition is also analyzed in this study. Figure 20 shows the comparison of the profiles in the target case and the final case, and the final profile is almost superimposed on the target one, which means that the stiffness of the obtained layup of the compliant skin is sufficient in takeoff condition. The target profile is obtained through aerodynamic optimization code under the cruising Mach number of 0.85, while the final one is extracted from the FEM results of the adjusted layups with the same flight condition. To simulate the locked condition of the morphing leading edge, the inner four interface points are applied with fixed boundary condition. It can be seen from Figure 21 that these two profiles fit perfectly

with a maximum deviation of 0.2250 mm, which means the designed compliant skin has sufficient ability to withstand the cruising aerodynamic force.

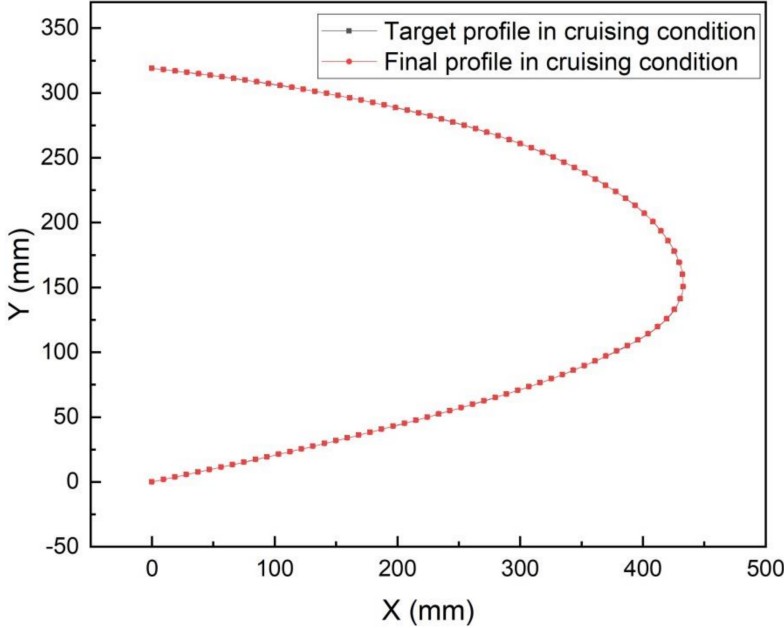

**Figure 20.** Comparison of the profiles in the target case and the final case.

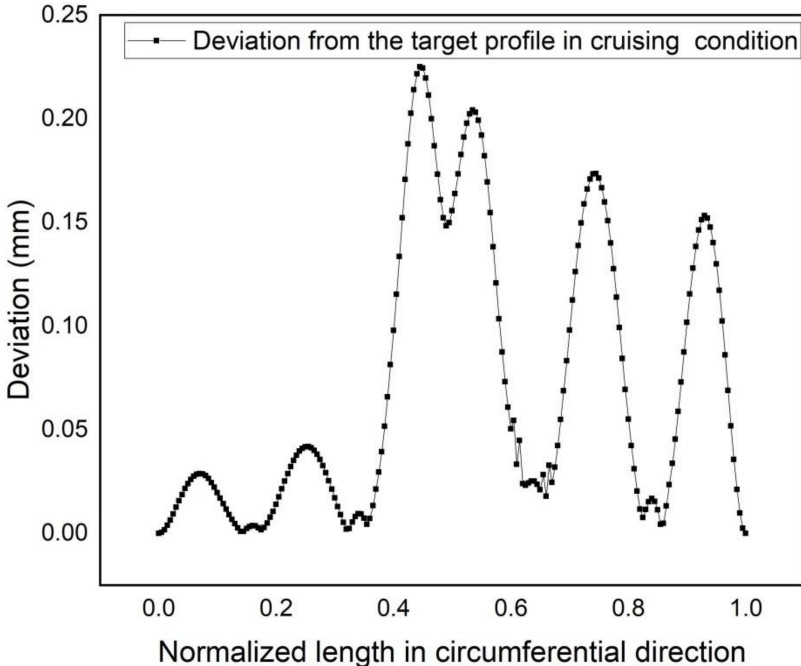

**Figure 21.** Deviation of the final profile from the target one in cruising flight condition.

### 6.2. Strength Assessment

Strength is the other significant factor for complaint skin design, especially for large-scale aircraft, and it is also assessed in the drooped case and the cruising case. As shown in Figures 22 and 23, the maximum values of tensile strain are 13,970 and 4184 μ, while the maximum values of compressive strength are 2321 and 751 μ, respectively. As illustrated in Section 3, the tensile ultimate strain and the compressive ultimate strain are 33,166 and 13,538 μ for the selected fiber-reinforced prepreg, which demonstrates that the structural strength of the compliant skin is sufficient, with the safety factors of 2.37 and 3.2.

Moreover, the maximum strain is distributed around the leading edge tip in the drooped case and around the upper surface for the cruising case. This can be explained by the maximum curvature change occurring around the leading tip in the drooped case and the maximum aerodynamic pressure being distributed on the upper surface.

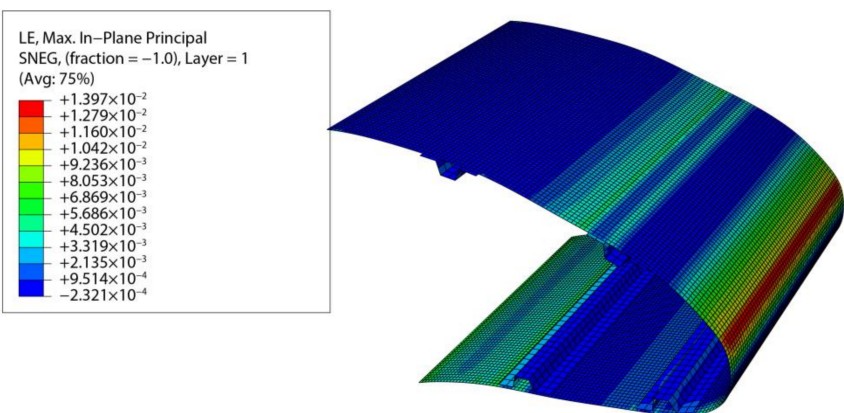

**Figure 22.** Strain contour of the compliant skin in drooped case.

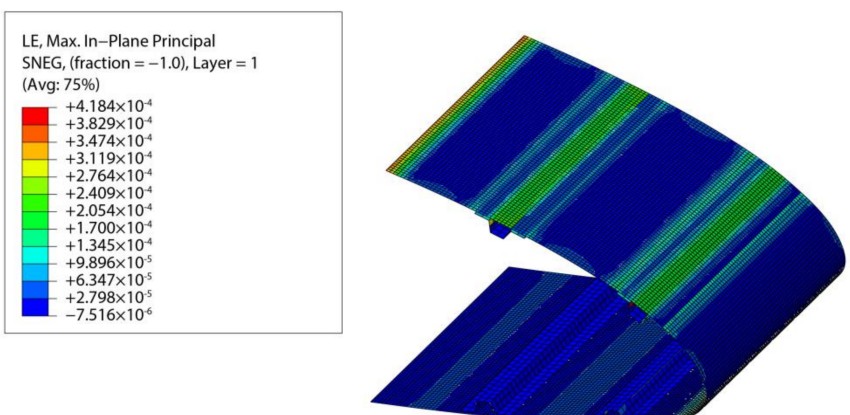

**Figure 23.** Strain contour of the compliant skin in cruising case.

## 7. Conclusions

This paper presents an innovative two-step design method for variable-stiffness complaint skin. Through rationally predefining a layup sequence in the first step and performing engineering adjustment of the optimized layups in the second step, the method can increase the deformation accuracy and simultaneously consider the manufacturing constraints. For the proposed design frame, this study also develops an improved objective function, WLSE, and employs a globally convergent optimization method, NSGA-II, to further ensure high deformation accuracy. The shape accuracy analysis results show that the obtained accuracy is good enough for a large-scale aircraft, and the deviation introduced by the layup adjustment process is negligible, yet it still needs to be improved in the future by, for example, using a single-step optimization method integrating manufacturing constraints directly to decrease deformation error. In addition, the strength assessment shows that the final maximum tensile and compression strains meet the engineering requirement of the safety factor.

**Author Contributions:** Conceptualization, Z.W.; methodology, Z.W.; software, Z.W.; validation, Z.W.; formal analysis, Y.Y.; investigation, Z.W.; resources, Y.Y.; data curation, Z.W.; writing—original draft preparation, Z.W.; writing—review and editing, Y.Y. All authors have read and agreed to the published version of the manuscript.

**Funding:** This research received no external funding.

**Institutional Review Board Statement:** Not applicable.

**Informed Consent Statement:** Not applicable.

**Data Availability Statement:** The data presented in this study are available in this article.

**Acknowledgments:** We acknowledge the strong support received from Jun Hua and Min Zhong from CAE. We thank Xiaopin Zhong from Northwest Polytechnic University for teaching us the NSGA II programming and Markus Kintscher from DLR for helpful discussions on the optimization.

**Conflicts of Interest:** The authors declare no conflict of interest.

## Nomenclature

| | |
|---|---|
| $d_i$ | Deformation value of the $i$th control point |
| $f_{xi}, f_{yi}$ | Actuating load exerted on the $i$th stringer in x and y directions |
| $(f_{xi}^L, f_{xi}^U)$, $(f_{yi}^L, f_{yi}^U)$ | Lower and upper load limit of the $i$th stringer in x and y directions |
| $LSE$ | Least square error |
| $n$ | Number of the control points |
| $s$ | The normalized length along the circumferential direction of the leading edge profile |
| $t$ | Maximum allowable thickness |
| $t(s)$ | Maximum thickness along the circumferential direction of the leading edge profile |
| $w$ | Weighted factor |
| $WLSE$ | Weighted least square error |
| $x$ | x component of the coordinate of every control point |
| $y$ | y component of the coordinate of every control point |
| $\varepsilon_{lim}$ | Material limit in terms of strain |
| $\Delta\kappa$ | Curvature variation |

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
