# Peer review of "Design of a Variable-Stiffness Compliant Skin for a Morphing Leading Edge"

_applsci, doi:10.3390/app11073165_

Round 1
Reviewer 1 Report
The paper details the design process of a variable-stiffness compliant skin structure to be used in a morphing leading edge of a commercial jet aircraft. Some points that need to be addressed before publication can be found below:
a) The significance and relevance of the research presented here is not clearly established in the paper. The heavy reliance on past work by Kintscher (7 out of the 24 references) is one thing; the lack of any explanation of the relevance of the present work is another. Is this a corollary of past work by others, and hence derivative and incremental, or is there something new and noteworthy here?
b) The quality of references reflects upon the present work as well: referencing intractable sources like minor conferences, and then basin the present work upon such foundation, treating it like absolute truth, and following blindly upon the path laid by the past work, all these do not inspire confidence in the present work. At the very least all the references should be traceable (DOI or other mechanism) so that a reader can evaluate the claims put forth by the authors, particularly since the validity of the present work rests heavily upon the past work.
c) All crucial aspects of the research methodology are missing from the present paper. Terms like "aerodynamic target profile" are used without any explanation, even though the concept is central in the present research. A lot more explanation is required about the aerodynamic code used by the author, and a lot of justification of statements regarding optimization. Once the details of the code are presented, validation and verification of the code will become an issue. A similar consideration applies to the FEA (?) code used for the modeling of the lay up; which code was used, what V&V took place, why do we trust the results, how are the objective functions defined, etc., are all questions. And the same applies with the NSGA methodology which is never explained, justified, or verified.
At present the paper does one thing well: it discusses the lay up configurations, the various constraints, and the optimization process related to the manufacturing. But the main issue of why is there an optimal aerodynamic profile, how are the various objective functions defined, how are any of the computations carried out, all that is currently missing.
Reviewer 2 Report
A clear and well-structured paper on a relevant topic, I have edited my few comments in the attached document.

Reviewer 3 Report
1.Nomenclature for all parameters in the paper needs to be sorted out.
2. Fig.3 shows the spanwise is 3000mm, but in the paper it is written as spanwise length of 350mm?
3. Please provide some reference about the target profile, why the arthors select this profile as the target profile.
Round 2
Reviewer 1 Report
The paper combines three different pieces of software in an iterative optimization procedure with the objective of "optimal" design of an airfoil morphing leading edge:
a) an aerodynamic code, developed by others and published in ref. 26, is used in order to come up with an "optimal" aerodynamic shape for the airfoil under examination. The code is a "black box" that is never explained, and the fitness, or not, of the generated profile is unknown, and unknowable. The reader gets no validation & verification for this code;
b) a genetic algorithm, developed by others and published in ref. 27, is used to conduct the optimization process, with "high efficiency" and "stability". The algorithm is another "black box" used during this study, without providing justification of the claims, or even basic confidence in the procedure. Test cases, examples where a noon-genetic algorithm was used and contrasted, a simple case where the answer is known beforehand, etc. would be needed here. Most importantly, the "improved objective function" used by the authors for optimization is completely 'ad hoc', stated in the paper without any justification whatsoever. Improved from what basline case, arrived at by which methodology, optimizing what exactly, all these are questions left ananswered;
c) a non-linear FEM code is used for the evaluation of properties in the variable stiffness lay-up. This is by far the least explained "black box" operation in the paper, with no references, and no validation & verification. What were the parameters of the calculations, was the code optimal for the calculation of laminates, or even capable? What is the limit of curvature that can be computed using the code, what is the meaning and importance of non-linearity of the code, etc. are all questions left unanswered.
The main issue with this work is that the research methodology is practically opaque, raising questions about the results stated in the paper. Where sufficient information is provided by the authors, it usually points to the work of others published elsewhere. There is scientific merit in the examination of the different laminates and lay-up procedures listed by the authors, but results like those presented in figures 18 and 20 are really obscured by the many questions raised by the obscure methodology.
Reviewer 3 Report
The authors have answered all questions, so the draft can be accepted in present form.
Author Response
For the revised manuscript, please see the attachment.

Round 3
Reviewer 1 Report
After a few round of review the authors have finally disclosed the nature of all three codes that they use in their study. None of those were developed in house, there is no validation or verification of any of their three pieces of software (aerodynamic code, optimization based on genetic-algorithm, FEA code). The authors do not appreciate that the research/industrial work of other authors do not automatically confer validity to their own derivative work. Every step of the research should be verified and justified before a journal article can be published with the results. All claims advanced in the work are left unsubstantiated, and resting on the quality of the work of others. This is not a scientific/engineering piece of work, it is simply a report of work in progress that would probably raise eyebrows even in the context of a design based graduate course.
Author Response
Dear Reviewer,
We have revised the manuscript. Actually, the innovation point of the paper is the in-house optimization framework proposed in this paper.
Would you please review it again? The updated manuscript is attached.
Thank you very much.
Best regards,
Zhigang
